## Research Article

sediment transport; remote sensing; machine learning; coastal monitoring; coastal waters

**Corresponding author:**
Aoife Igoe;
Email: igoea@tcd.ie

# Modelling suspended sediment concentration in coastal Ireland using machine learning

Aoife Igoe[1] ⓘ, Iris Möller[2] ⓘ and Biswajit Basu[3]

[1]Department of Electronic and Electrical Engineering, Trinity College Dublin, Dublin, Ireland; [2]Department of Geography, Trinity College Dublin, Dublin, Ireland and [3]Department of Civil, Structural and Environmental Engineering, Trinity College Dublin, Dublin, Ireland

## Abstract

Coastal environments are highly dynamic, making monitoring of suspended sediment concentration (SSC) both challenging and essential. SSC serves as an indicator of coastal processes, storm impact, water quality and ecosystem service delivery. However, direct measurement of SSC is costly, logistically difficult and spatially limited. Although remote sensing offers a promising alternative by estimating SSC from surface reflectance, it requires calibration and is often constrained by site-specific applicability. This study presents a machine learning framework for national-scale SSC estimation using Landsat-8 and Sentinel-2 imagery, calibrated with 147 in situ SSC samples. Several models were evaluated, with XGBoost yielding the best performance ($R^2 = 0.72$, RMSE = 17 mg/L). SHapley Additive exPlanations values were used for model interpretability. Visible and infrared bands, along with geographic features, were identified as key predictors, reflecting the importance of coastal typology in shaping the SSC-reflectance relationship. The model's value was demonstrated through a 10-year spatio-temporal analysis of SSC in Wexford Harbour. Seasonal patterns showed higher estuarine mixing in winter, while high SSC events coincided with rainfall and strong winds, indicating responsiveness to meteorological drivers. These findings highlight the potential of integrating remote sensing and machine learning for scalable, interpretable and cost-effective SSC monitoring.

## Impact statement

Climate change and land-use change are threatening the functioning and quality of coastal environments in Ireland, as elsewhere across the globe. Suspended sediment concentration in coastal waters acts as an indicator of coastal dynamics, storm impact, water quality and ecosystem service delivery. Its measurement is thus of extreme importance to coastal management and land-use planning, and capturing temporal and spatial fluctuations in suspended sediment concentrations is critical for informed environmental management and decision-making. Measuring SSC is also notoriously difficult, as direct sampling of coastal waters is at best costly and at worst impossible, compromising the ability of governments and public agencies to monitor SSC. Remote sensing from aircraft or satellites allows us to estimate SSC remotely but this has other challenges, such as cloud cover or the complex way in which many constituents of coastal water (e.g., algae) reflect sunlight and complicate the SSC 'signal'. We offer a methodology for estimating SSC in the coastal waters of Ireland using machine learning. As there are some direct measurements within Irish coastal areas (from water samples largely collected to meet Ireland's obligations as part of the EU's Water Framework Directive's), we were able to compare measured with remotely estimated SSC using a combination of NASA's Landsat-8 and Copernicus Sentinel-2 satellite imagery. As the relationship between actual and satellite estimated SSC is heavily affected by the type of coast, we see an influence of geographic location on the model developed. The resultant machine-learning tool has the advantage that it can be continuously improved as more satellite imagery is acquired, with minimal field sampling effort. If adopted by governments and public agencies as a tool to monitor SSC, spatially explicit coastal management and planning will improve markedly.

## Introduction

Suspended sediment concentration (SSC) is an important parameter to monitor at the coast. Changes in SSC can reflect coastal erosion and affect the formation of coastal landforms, as well as impacting how coastal landforms persist and continue to provide coastal flood protection. Coastal wetland areas are particularly sensitive to changes in SSC. Within shallow estuarine settings allochthonous (externally derived and tidally imported) sediment has been shown to be a critical determinant of an individual coastal wetland's ability to accrete upwards (French et al., 1995). Once compaction and shallow subsidence has been taken into account (see, e.g., Allen

(2000)), such accumulation determines the wetland's elevation relative to sea-level rise. Under conditions of low wave energy, suspended sediment can also deposit on tidal flats and influence the time to maturity of salt marshes or mangroves, which provide many important ecosystem services (Lovelock, 2008; Currin et al., 2017). Thus, in addition to requiring sufficient accommodation space (e.g., landwards migration), whether intertidal wetlands can persist in the face of a rise in sea level is critically determined by SSCs (see also Saintilan et al. (2022) and Kirwan and Megonigal (2013)). Sediment in tidal waters also plays an important role in impacting water quality and primary production, both of which are key controls on the shallow-water marine food web (Bilotta and Brazier, 2008). Spatial patterns and temporal changes in SSC are thus important in affecting recreational and commercial marine fisheries.

Importantly, recent global climatic and regional and local land-use changes have led to changes in many of the controls on sediment delivery and distribution in shallow coastal seas. Although land-use changes such as dam construction, river dredging and flood defences have significantly altered the release of sediment from river catchments (Syvitski et al., 2005; Heritage and Entwistle, 2020), there has also been an increasing intensity of meteorologically induced storm surges (Debernard et al., 2002; Michaels et al., 2006), and changes in the behaviour of sediment (e.g., the flocculation of clay particles, which is dependent on salinity and flow velocities (Mietta et al., 2009) in the coastal ocean. The spatial distribution of SSC is thus of particular interest in areas that have coastlines vulnerable to flooding or erosion and dependent on the deposition and configuration of the shallow intertidal zone. In Ireland, such areas include Wexford Harbour. In such locations, better monitoring of SSC can aid in planning of adjacent land-use and coastal flood risk management. Current modelling of SSC, however, is often based on point measurements at specific locations for water quality assessment, at long and irregular time intervals. Knowledge on the spatio-temporal patterns of SSC is thus limited by the spatial distribution of the sampling sites, which does not allow for sufficient frequency of observations over larger ($\geq$km$^2$) areas and time periods ($\geq$decades).

### Remote sensing of SSC

Remote sensing has become a powerful tool for monitoring inland and coastal water bodies. Earth observation satellites, such as those in the Landsat, Sentinel and MODIS missions, acquire imagery across a range of spectral bands, from visible to near-infrared and shortwave infrared, allowing for consistent large-scale observations of surface conditions. These sensors measure top-of-the-atmosphere radiance, which is processed to yield surface reflectance: the proportion of incoming solar radiation reflected by the Earth's surface back towards the sensor at different wavelengths (Wang et al., 2020).

In aquatic environments, surface reflectance is influenced by the optical properties of the water column, which are, in turn, affected by various constituents, including suspended sediments, coloured dissolved organic matter (CDOM), phytoplankton (quantified via chlorophyll a) and dissolved substances (Gholizadeh et al., 2016). SSC, in particular, plays a dominant role in modulating water-leaving reflectance, primarily through the scattering and absorption of light. Because suspended particles alter the reflectance signature in specific spectral regions it is possible to relate satellite-derived surface reflectance to SSC using a range of modelling approaches.

Analytical and semi-analytical methods require detailed information about the water column, including depth, sediment characteristics (e.g., mass, rock type and grain size) and the relative proportions of CDOM and SSC (Wang et al., 2020). Montanher and de Souza Filho (2015) found that different spectral bands were needed for modelling SSC, depending on whether the water was dominated by inorganic particles or a combination of inorganic and phytoplankton. The turbidity of the water also affects the best spectral bands for modelling (Gholizadeh et al., 2016). These methods necessitate comprehensive local water studies, making the resulting models highly location-specific. Empirical methods, by contrast, rely primarily on SSC samples collected near the time of satellite image capture. These samples are used to establish a statistical relationship between surface reflectance and SSC (Wang et al., 2020). Several challenges arise when using these methods. First, they often remain location-specific, as the relationship between reflectance and SSC is influenced by the particular particulate matter present, as well as water depth. Second, these methods require a substantial number of SSC samples collected concurrently with satellite overpasses under cloud-free conditions, particularly for dynamic areas.

Research on coastal SSC modelling has primarily focused on location-specific empirical models, often achieving good results in non-turbid waters (<100 mg/L) using multiple spectral bands. However, in turbid waters, model performance frequently deteriorates, likely due to reflectance saturation in visible bands around 100 mg/L and in non-visible bands between 500 and 1,000 mg/L (Luo et al., 2018). As a result, remote sensing, based solely on surface reflectance, becomes less effective for detailed SSC modelling in highly turbid waters (Shahzad et al., 2018).

Given the prevalence of local-specific models, most studies either target highly turbid waters, such as rivers, or waters with low turbidity (Marinho et al., 2021). One of the major challenges in applying remote sensing to SSC modelling is obtaining a sufficiently large and representative dataset of in situ SSC samples for calibration. This is particularly critical in coastal regions, which often experience high spatial and temporal variabilities in SSC and are vulnerable to processes on instantaneous timescales, such as localised erosion, that can have a high but potentially short-lived impact on sediment in the water column. Identifying and quantifying these changes is essential for effective management and mitigation strategies.

### Machine learning models

Traditional approaches for SSC modelling in the literature often rely on regression models using one or more spectral bands (Knaeps et al., 2015). These models have used various regression forms, including linear, log-linear and polynomial equations, to relate surface reflectance to SSC. Although relatively simple and interpretable, such models are typically limited in their ability to capture complex, non-linear relationships and often require location-specific calibration. To address these limitations, more recent studies have explored machine learning (ML) techniques, including Random Forests and gradient boosting methods, which offer enhanced predictive capabilities. For instance, Hu et al. (2023) combined spectral bands with weather and river flow data to estimate monthly SSC using a gradient boosting model in the lower Yellow River in China. ML models have become increasingly popular in the study of coastal sediment transport (Goldstein et al., 2019), driven by the growing availability of remote sensing and environmental data.

However, ML models also present significant challenges. Chief among these is their reliance on large, high-quality training datasets. Without sufficient data, especially labelled SSC samples, models are prone to overfitting and poor generalisation (Goldstein et al., 2019; Brigato and Iocchi, 2021). This leads to overconfidence in the model and low performance outside the training dataset. Deep learning models, such as neural networks, are particularly data-intensive and have seen limited application due to the high cost and logistical complexity of acquiring adequate in situ samples.

Interpretability remains a key concern when applying ML in environmental sciences. SHapley Additive exPlanations (SHAP) has emerged as a widely used method for interpreting complex models. Rooted in game theory (Shapley et al., 1953), SHAP treats each feature as a player in a cooperative game and allocates the model's output to features based on their marginal contributions. It provides local explanations that show how individual input features influence model predictions. SHAP is especially effective for explaining ensemble models like Random Forests and XGBoost, which otherwise would be a black box, by looking at the importance of the features across the ensemble, making it more stable for ensemble methods than sensitivity analysis. It has been successfully applied in environmental modelling for feature selection, model transparency and diagnostics (Lundberg and Lee, 2017; Tang et al., 2022).

The primary goal of this study was to develop a model capable of capturing spatio-temporal patterns of SSC to gain insights into the dynamic nature of SSC in coastal waters, taking advantage of the spatio-temporal coverage of satellite-based remotes sensing. Information on such patterns and their dynamics over time is needed both for furthering our marine and coastal ecological and geomorphological knowledge base but also for tailoring land and coastal management practices in a way that allows adaptation to climatic change and mitigation of climate change impacts. The advantages of the model's ability to accurately detect patterns and changes in SSC, its sensitivity to variations thus outweigh the fact that its ability to exactly predict SSC at any given point in place and time is necessarily limited.

## Materials and methods

### Data

#### Satellite imagery

This study used imagery from the Harmonised Landsat and Sentinel-2 (HLS) dataset, developed by NASA to provide consistent surface reflectance products from Landsat-8/9 (OLI) and Sentinel-2A/B (MSI) satellites (Claverie et al., 2018). By harmonising bandpass differences, spatial resolution (30 m) and applying bidirectional-reflectance-distribution-function normalisation, the dataset enables high temporal resolution (2–3 days) through combined satellite observations. The satellite images were obtained and processed using Google Earth Engine (Gorelick et al., 2017).

#### SSC data

In situ SSC samples were obtained from the EPA and Eden Ireland, covering the period 1992–2024, collected as part of the Water Framework Directive's monitoring of transitional and coastal waters (Environmental Protection Agency, 2024). Only surface and grab samples were included because the spectral signal weakens with depth (Curran and Novo, 1988). Each sample was taken at a monitoring station, which had a unique set of coordinates.

#### Combined dataset

In order to use remote sensing imagery as input to an SSC model, calibration to the study area is needed. This requires a set of samples matched with satellite images within a short time period, or overpass. The number of days between sample measurement and satellite image capture, and the timing of the sample, are particularly important in coastal areas, where there is a high amount of change on short timescales, and where the timescale and degree of such change is itself time-dependent (e.g., seasonally variable). It is thus to be expected that the accuracy of any model is improved where samples are collected as close as possible in time to the time of satellite overpass. Unfortunately, this is particularly tricky in areas that receive a lot of clouds and precipitation, such as coastal regions of Ireland, and can limit the amount of available data. This study uses a strict overpass of ≤1 day, which allows for a suitable range of SSC values to be used for calibration, with 151 samples available in total. Similar studies such as Yepez et al. (2018), which modelled SSC in the range of 18–203 mg/L used an overpass of 1 day, while Dethier et al. (2020) tested an overpass range of 0–8 days and found that 2 days best balanced accuracy with uncertainty for their study area. The location of each monitoring station, with the number of samples available is shown in Figure 1A, and a histogram of the SSC in a log scale is shown in Figure 1B. There were 147 in situ samples that were matched with satellite images, from 78 unique monitoring stations between July 2013 to October 2024. Ninety-seven of the images were from Landsat-8 and 50 were from Sentinel-2.

### Methods

This study involved data pre-processing, data aggregation and comparing modelling methods for prediction and validation of SSC. The code used to produce the results in this article is publicly available to download on the authors GitHub repository: https://github.com/igoea20/Remote_Sensing_SSC_Ireland.

#### Data pre-processing

Remotely sensed spectral data require a high-amount of pre-processing to ensure its accuracy, particularly in areas where there is a high amount of cloud cover, such as the Irish coast. Cloud and shadow masking was performed using the Fmask quality bands, masking cirrus, cloud, cloud shadow and cloud-adjacent pixels based on the approach described by Qiu et al. (2019). Known limitations of the S30 cloud detection are addressed using a time-series outlier-filtering method adapted from Chen and Guestrin (2016), which applies a Hampel filter and temporal-consistency analysis using the modified Normalised Difference Water Index (mNDWI), which is a ratio of the green (0.53–0.59 μm) and shortwave-infrared (1.57–1.65 μm) bands (Vermote et al., 2008; Claverie et al., 2018). Cloud-contaminated or physically implausible values (e.g., negative reflectance) were removed. Water pixels were identified using the mNDWI (Xu, 2006).

For the in situ samples of SSC, some data points had to be removed due to their unsuitability to remote sensing. Measurements from water shallower than 1 m were excluded to reduce errors from sediment bed backscattering. Only samples from depths ≤5 m were used to ensure that the satellite-derived signal corresponded to the upper water column, as the penetration reduces with turbidity (Curran and Novo, 1988).

#### Random forest

Random Forest regression, an ensemble method based on decision trees, was implemented using Scikit-learn (Pedregosa et al., 2011).

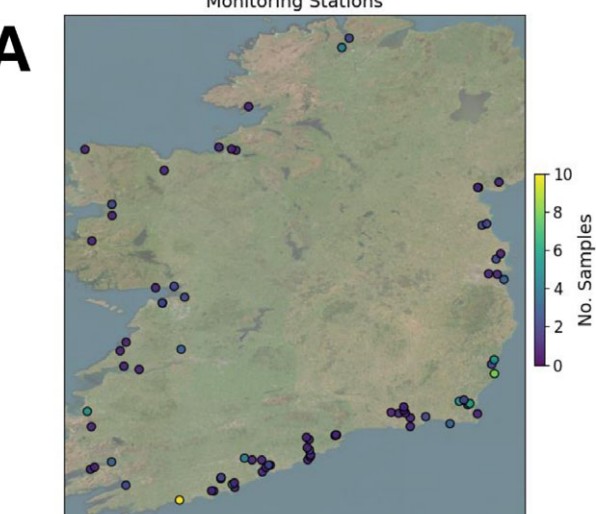

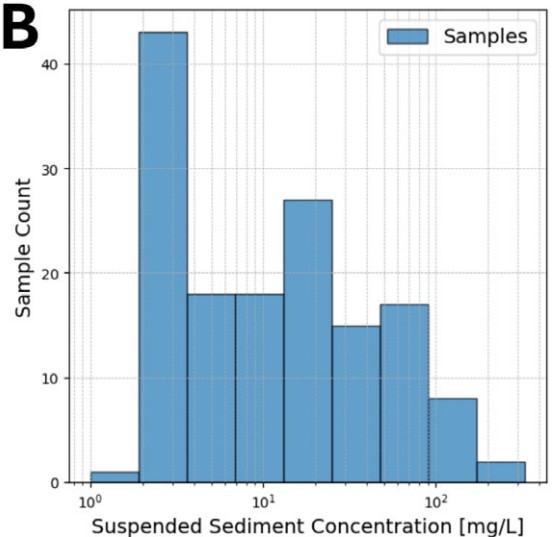

**Figure 1.** Location and distribution of the sampled SSC. The locations of the monitoring stations and the number of samples from each station are shown in (A), with the distribution (in the log scale) shown in (B).

It uses bootstrap samples to train individual trees, with predictions averaged to improve accuracy and reduce overfitting. To use RF models, it is necessary to adjust the model's hyperparameters to suit the data and problem in question. RandomizedSearchCV was used to randomly search a grid of hyperparameters, choosing the optimal hyperparameters that minimised root mean squared error (RMSE). The optimal hyperparameters found were as follows: number estimators of 50, min samples in a split of 2, min samples in a leaf of 1, max features of 1 and max depth of 7.

### Extreme gradient boosting

XGBoost (Chen and Guestrin, 2016), a gradient boosting framework, builds sequential models where each minimises the errors of its predecessor, with the model consisting of many weak learners (small regression models), and the final predictions being the weighted sum of the predictions from the weak learners. It has improved control against overfitting compared to Random Forest through regularisation. The XGBoost library (version 2.1.2) was used (Chen et al., 2016), with hyperparameters tuned using RandomizedSearchCV. The optimal hyperparameters found were as

follows: number of trees of 100, tree depth of 4, learning rate of 0.03 and subsample of 0.7. To improve the model interpretability, SHAP values were computed for the final XGBoost model, allowing insight into feature contributions and reducing its "black-box" nature.

### Multi-layer-perceptron

Multi-layer perceptron (MLP) is a simple form of feedforward artificial neural network, and was implemented using Scikit-learn (Pedregosa et al., 2011). Due to the limited number of samples available for training, it was configured with one hidden layer. Hyperparameters, such as the number of neurons in the hidden layer, learning rate and regularisation strength, were optimised using RandomizedSearchCV. The optimal hyperparameters found were as follows: solver = 'adam', initial learning rate = 0.03, hidden layer size = 10, alpha = 0.01 and activation = 'relu'.

### Input variables

Input features to the model included the spectral bands, band ratios and spatial coordinates. The coordinates were included to account for regional environmental gradients and potential spatial autocorrelation. The input vector was as follows: ['Blue', 'Red', 'Green', 'NIR Narrow', 'Blue/Red', 'Blue/Green', 'Red/Green', 'SWIR 1', 'Latitude', 'Longitude'], where Blue (0.45–0.51 μm), Red (0.64–0.67 μm) and Green (0.53–0.59 μm) are the visible bands, NIR narrow (0.85–0.88 μm) is the near-infrared band and SWIR 1(1.57–1.65 μm) is the Shortwave Infrared band.

### Model evaluation

Model performance was evaluated using leave-one-out cross-validation (LOOCV) (Hastie et al., 2005). In this approach, the dataset of size N is split into N iterations, each using $N-1$ samples for training and the remaining one for testing. This method ensures each data point is tested once, providing an unbiased estimate of model generalisation, and ensuring the performance is reflective of the whole dataset. Model performance was evaluated using the RMSE (Equation 1), the coefficient of determination ($R^2$, Equation 2) and the relative percentage bias (Equation 3), where $SSC_i$ is the true in situ value of SSC for observation $i$, $S\hat{S}C_i$ is the predicted value of SSC for observation $i$, $\overline{SSC}$ is the mean value of observed SSC and $n$ is the total number of observations.

$$\text{RMSE} = \sqrt{\frac{\sum \left(SSC_i - S\hat{S}C_i\right)^2}{n}} \qquad (1)$$

$$R^2 = 1 - \frac{\sum \left(SSC_i - S\hat{S}C_i\right)^2}{\sum \left(SSC_i - \overline{SSC}\right)} \qquad (2)$$

$$\text{Rel.Bias} = 100 \times \frac{\frac{1}{n}\sum \left(S\hat{S}C_i - SSC_i\right)}{\overline{SSC}} \qquad (3)$$

## Results

### Model performance

The results for all three modelling approaches are shown in Table 1. The XGBoost method demonstrated the highest model performance with $R^2 = 0.72$, RMSE = 17 mg/L, Rel Bias = $-1.8\%$. The scatter plot in Figure 2A) shows the results from the LOOCV predictions, compared to the in situ samples. Overall, the model was able to

**Table 1.** Results from LOOCV of the machine learning models

| Model | RMSE [mg/L] | $R^2$ | Rel. bias (%) |
|---|---|---|---|
| Random Forest | 19 | 0.65 | −0.68 |
| MLP | 23 | 0.47 | 2.77 |
| XGBoost | 17 | 0.72 | −1.8 |

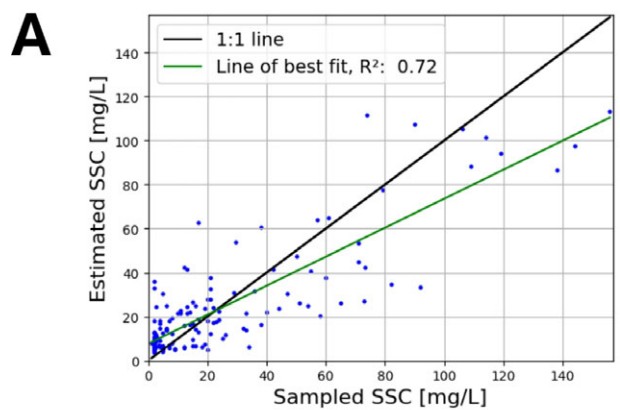

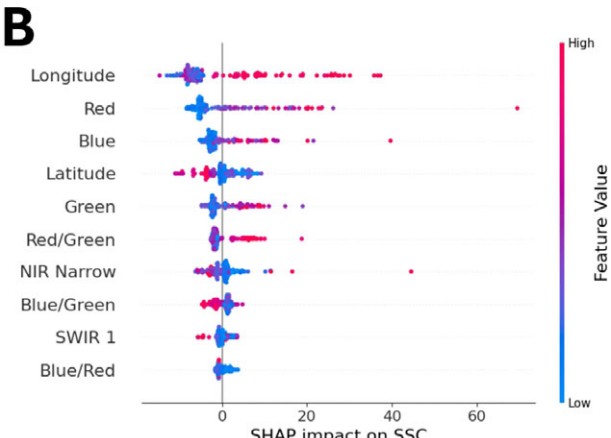

**Figure 2.** (A) The modelled SSC, using the XGBoost model, is shown in blue. Each point is from an LOOCV iteration. The green line shows a linear regression between observed and predicted SSC. (B) The SHAP analysis of the input features is shown with the x-axis showing whether the feature increased or decreased SSC. The colour bar indicates whether the sample had a high or low value for that feature.

learn the distribution, but there was a lot of scatter around the y = x line.

### *Feature importance*

Figure 2B shows the SHAP summary plot of the XGBoost model, indicating the impact of each feature on the SSC output. The x-axis shows the SHAP value of each feature, with a value >0 indicating that the feature pushed the prediction higher, and a value <0 means the feature lowered the predicted SSC. The colour of each point indicates whether the feature value was high or low. Each point indicates a training point in the model. Longitude is shown to have the largest overall impact on model predictions, with higher values (the east of the country), tending to increase SSC. This suggests that regional differences, such as contrasting geology, sedimentology and glacial history, as well as exposure to the predominant westerly airflow, strongly influence SSC, and we can see that there is a non-

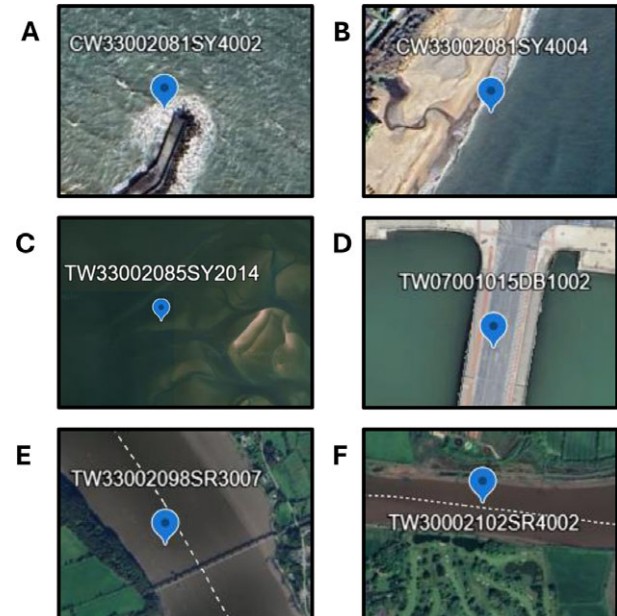

**Figure 3.** Six monitoring stations were identified that could not be accurately predicted using the model.

linear relationship, as expected (Devoy et al., 2021). The red and blue bands both have significant influence on SSC, with lower red or blue values tending to decrease SSC. Latitude is less important, but we can see that there is an indication of north–south differences, with higher latitude tending to decrease SSC. The other bands (non-visible NIR narrow and SWIR 1, and band ratios) have less of an impact on SSC, and they tend to show complex relationships with SSC, due to the relationship being non-linear. We see that a high Blue/Green is associated with lower SSC (lower turbidity). A combination of short and long wavelengths takes advantage of deeper water penetration and sensitivity to high values of SSC (Curran and Novo, 1988).

Several monitoring stations had consistently high prediction error (>20 mg/L); some of these locations are shown in Figure 3. The errors at the monitoring stations can be explained as follows: in (A), there is wave breaking and diffraction around a man-made structure; in (B), there is shallow-water wave shoaling; in (C), it is a shallow subtidal area with surface reflectance of the bed changing between low and high tides (spring tidal range of 1.5 m, neap of 0.9 m; Hartnett and Nash, 2004); in (D), there is an artificial surface above the water body and in (E) and (F), there are tidal inner-estuary channels.

### *Seasonal- and event-based patterns in SSC*

The developed model facilitates investigation of both seasonal variations and event-driven anomalies in SSC. Figure 4 illustrates the seasonal distribution of SSC within Wexford Harbour, comparing the winter period (December 2022–February 2023) with the summer period (June 2023–August 2023).

Figure 5 provides additional insight into potential environmental drivers of extreme SSC events. Figure 5A displays the monthly distribution of daily total rainfall and average windspeed measured at Johnstown Castle in Wexford over the period 2014–2024. Superimposed red lines indicate years in which SSC exceeded 140 mg/L, highlighting the temporal alignment between extreme SSC values

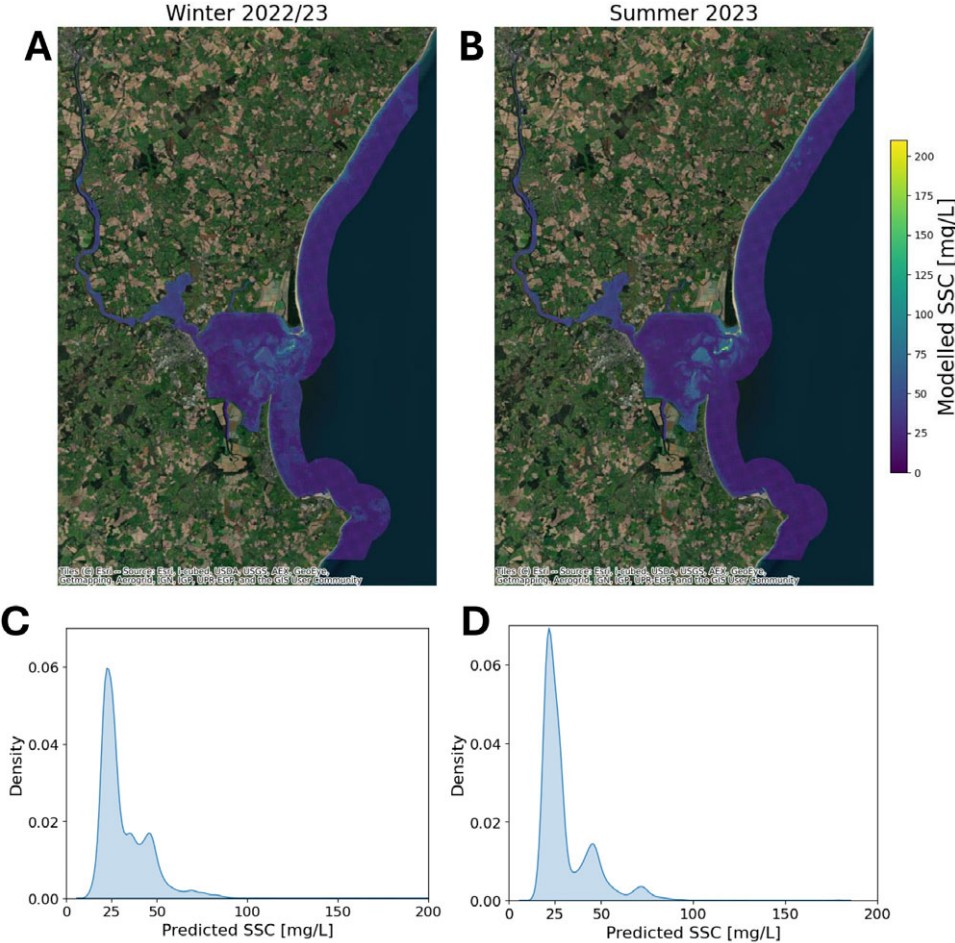

**Figure 4.** The seasonal median SSC is shown for Wexford Harbour. (A) The SSC from December 2022 to February 2023. (B) The SSC from June 2023 to August 2023. The distribution of SSC for (A) is shown in (C), and the distribution of (B) is shown in (D).

and weather extremes. Between 2014 and 2025, eight SSC measurements exceeded 140 mg/L, spanning five unique dates: 03/10/2019, 19/10/2022, 08/07/2023, 27/09/2023 and 13/06/2024. These events were cross-examined against concurrent meteorological conditions. Notably, the SSC peak in June 2024 coincided with anomalously high daily rainfall for that month, as observed in Figure 5B. Similarly, high-rainfall conditions were also observed during the SSC peaks in September 2023 and October 2022, Figure 5C shows that the SSC events on 27/09/2023 and 03/10/2019 corresponded to days with unusually high windspeed for those months.

## Discussion

The XGBoost model had the highest $R^2$ value and lowest RMSE, and was chosen as the best of the ML models tested for remotely sensed SSC in coastal Ireland. Feature attribution using SHAP analysis provided additional insights into the model's behaviour. Among the input features, longitude was more influential than latitude, indicating a pronounced east–west spatial gradient in the SSC–spectral reflectance relationship. This spatial dependency is likely due to differences in coastal geomorphology, hydrodynamics and sediment characteristics between the Irish Sea and Atlantic-facing coasts, and exposure to the predominant westerly airflow (Gallagher et al., 2014), (Devoy, 2008). SHAP analysis also

confirmed that the visible bands, particularly blue, green and red, were among the most important spectral features.

### Interpreting trends in SSC

In Figure 4, a clear seasonal signal is evident, with more mixing in the winter months. Although the median SSC for the whole estuary is similar (32 mg/L for winter and 31 mg/L for summer), the spatial distribution of SSC is different as observed in Figure 4C and D. In summer, 70% of the pixels are less than 30 mg/L, compared to 60% in winter. The maximum SSC in winter is 209 mg/L in winter and 179 mg/L in summer. This pattern of elevated SSC in a wider spatial area may be attributed to increased hydrodynamic activity, including higher river discharge and wind-driven resuspension during the winter season. Bowers et al. (1998) identified strong seasonal variations in suspended sediment in the Irish Sea.

The model also facilitates the identification and analysis of extreme suspended SSC events, as illustrated in Figure 5. When examined alongside concurrent meteorological data, including daily total rainfall and average windspeed, these high-SSC episodes frequently coincide with periods of intense weather activity. In the Wexford Harbour case study, six remote sensing detected SSC peaks were investigated. Of these, three were associated with anomalously high monthly rainfall, while four corresponded with elevated wind speeds. These observations are consistent with previous

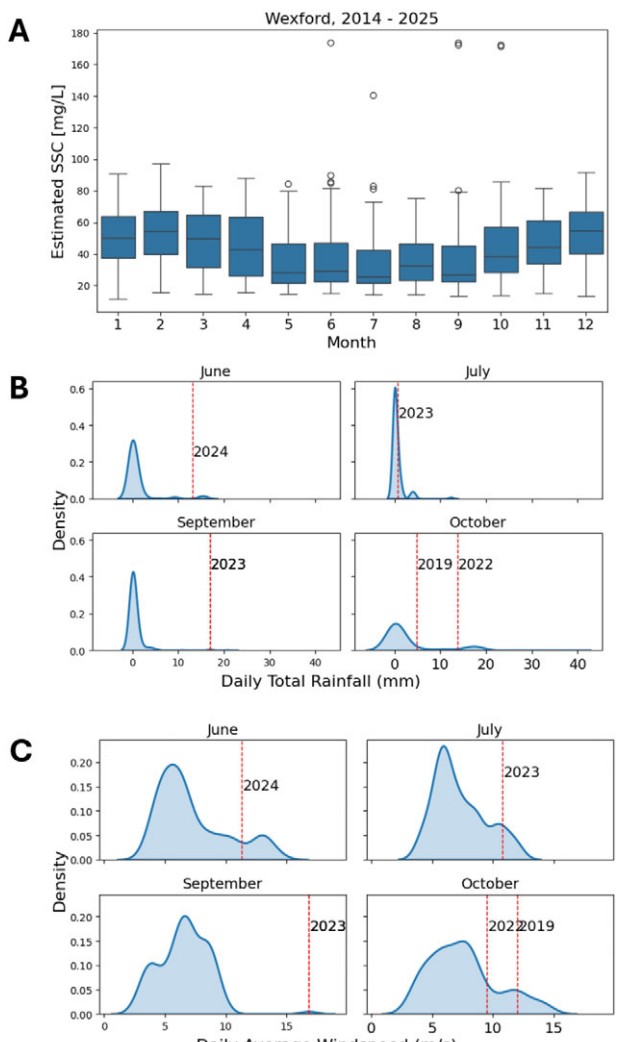

**Figure 5.** A) The monthly distribution of daily total rainfall measured at Johnstown Castle in Wexford. The red lines mark the years that had SSC values over 140 mg/L in that month. B) The monthly distribution of daily average windspeed measured at Johnstown Castle in Wexford. The red lines mark the years that had SSC values over 140 mg/L in that month.

findings suggesting that both runoff and wind-driven resuspension significantly influence episodic increases in SSC (Kalnejais et al., 2007; Drewry et al., 2009). Fluvial input, in particular, emerges as a likely contributor to such events, while windspeed appears to play an additional role in mobilising and resuspending sediments, further elevating SSC levels. Further research, with additional data for a greater set of extreme events could allow for a better understanding of the drivers of SSC and whether it is from runoff or wind-driven resuspension. To understand this relationship from a causal standpoint, we suggest further development of methodology.

Meteorological records also indicate the occurrence of named storms in close temporal proximity to several of the identified SSC events. Notably, Storm Agnes occurred on 27 September 2023, coinciding with one of the highest SSC values observed during the study period. Similarly, Storm Lorenzo impacted the region on 4 October 2019, shortly after an SSC spike recorded on 3 October 2019 (Met Éireann, 2025). These temporal alignments reinforce the hypothesis that extreme weather events can act as significant triggers for abrupt increases in coastal SSC (Miller, 1999; Suursaar et al., 2015).

Collectively, these findings highlight the model's capability to capture both spatial and temporal variabilities in SSC. In addition to identifying high-SSC zones and seasonal trends, it proves effective in detecting episodic events linked to environmental drivers such as rainfall anomalies, storm activity and wind-induced resuspension.

### Study limitations and next steps

A key limitation of the model lies in its reduced accuracy at higher SSC (>75 mg/L) levels. This issue is evident in Figure 2 and is consistent with previous findings on reflectance saturation at elevated SSCs (Curran and Novo, 1988; Markert et al., 2018; Shahzad et al., 2018). Reflectance becomes less sensitive to additional suspended material beyond certain thresholds, particularly due to the optical saturation of visible and near-infrared bands (Bowers et al., 1998; Doxaran et al., 2002; Luo et al., 2018). Moreover, ML models such as XGBoost and Random Forest are inherently non-extrapolative, meaning their predictions are restricted to the range observed in the training data (Chen and Guestrin, 2016). Therefore, caution is needed when interpreting model outputs in high-turbidity regimes, and they should not be treated as absolute estimates outside the validated range. A major contributing factor to this limitation is the under-representation of high-SSC samples in the training dataset. Expanding the calibration dataset to better capture high-turbidity conditions would be a logical next step. Targeted field sampling in known high-turbidity areas, coordinated with satellite overpasses, could enhance the model's predictive power and ability to model extreme sediment conditions.

Figure 3 highlights several monitoring stations where SSC predictions were problematic. These cases emphasise the importance of quality control in calibration data and the need for manual inspection and filtering to ensure representativeness. Remote sensing models must also be applied cautiously, particularly in tidal areas where water depth fluctuates and may push pixels in and out of the valid range for SSC estimation (Pahlevan et al., 2017; Dethier et al., 2020).

The lack of high-resolution, up-to-date bathymetry data for Ireland's coastal waters presents an additional constraint (O'Toole et al., 2020). Without accurate bathymetric information, the reliability of reflectance-based SSC estimates diminishes in shallow or variable-depth regions. Addressing this will require improved tidal prediction tools and detailed bathymetric surveys to support broader operational use.

This study also raises broader questions around the complexity and interpretability of ML models in environmental science. Although achieving high predictive accuracy is important, it must not come at the expense of transparency and rigorous validation. This includes using cross-validation, multiple performance metrics and interpretability tools such as SHAP values. However, it is important to note that SHAP, while useful, only provides correlational insight. Moreover, model performance is constrained by the quality and size of the training data, requiring thoughtful choices around regularisation, architecture and parameter tuning—especially in deep learning models such as neural networks (Karpatne et al., 2018; Zhu et al., 2023).

Although results were visualised using downsampled outputs for clarity, the model retains its full 30 m spatial resolution, enabling fine-scale environmental monitoring in regions as small as 5 km². This makes the method particularly well-suited for event-based

studies (e.g., storms or floods), multi-year trend assessments and local-scale management decisions. For example, it can help evaluate post-construction sediment changes around coastal infrastructure (e.g., breakwaters or tidal barrages) by comparing recent SSC patterns to historical baselines. It also holds promise for the long-term monitoring of sediment-sensitive ecosystems such as estuaries, saltmarshes and wetlands.

In addition to expanding the dataset and improving bathymetry, future research could explore the use of causal inference methods to go beyond correlational models and gain a mechanistic understanding of the drivers of SSC variability. This could yield more actionable insights for environmental planning and policy, especially in coastal zones prone to rapid sediment changes.

## Conclusions

In this study, we developed and validated an ML approach for modelling SSC in coastal areas using remote sensing data, incorporating geographic information to improve predictive accuracy. Our model, based on XGBoost, integrated visible and infrared spectral bands from Landsat and Sentinel satellites with spatially explicit geographic data, and was rigorously evaluated using LOOCV.

The model effectively captured key spatio-temporal patterns of relative SSC in shallow coastal waters, demonstrating strong performance across multiple scales. At the regional level, it successfully identified SSC dynamics across thousands of kilometres surrounding the island of Ireland. At the local scale, its application to multi-temporal imagery of Wexford, Ireland, revealed seasonal and event-driven sediment patterns that were consistent with known meteorological, hydrodynamic and fluvial processes at that site. Wexford estuary is a drowned valley estuary with a barrier, with flood-tidal dominance. Sediment supply forming the sediment deposits is heavily impacted by seasonal tides and flooding, with a large internal fetch distance meaning that waves are generated that can resuspend SSC and modify the shoreline (Cooper, 2016).

Given the complexity and variability of Ireland's coastal zones, shaped by a range of environmental drivers, our findings are encouraging. They indicate that this modelling framework can accommodate location-specific dynamics within a unified and scalable SSC monitoring approach. Although further refinement is warranted, particularly through more sophisticated integration of geographic information, such as geographic regression techniques or spatial clustering of regions, our results highlight the potential of remote sensing-based SSC monitoring. Such methods can support local and national agencies in tracking sediment dynamics across seasonal to multi-annual timeframes and spatial scales ranging from tens of meters to the national level. Ultimately, this approach can inform adaptive land and coastal management strategies that promote ecological resilience, geomorphological stability and climate adaptation in dynamic coastal environments.

**Open peer review.** To view the open peer review materials for this article, please visit http://doi.org/10.1017/cft.2025.10016.

**Data availability statement.** The Landsat-HLS and Sentinel-HLS data (available online at) were accessed using Google Earth Engine and are freely available.

The processed satellite datasets and trained models are available from the corresponding author, A.I., upon reasonable request, excluding the original dataset of in situ samples provided by the Environmental Protection Agency.

The python scripts used for data pre-processing, model training and result visualisation are available at https://github.com/igoea20/Remote_Sensing_SSC_Ireland.

**Acknowledgements.** The authors acknowledge that raw station datasets from Met Éireann are published under Creative Commons Attribution 4.0 International (CC BY 4.0). (https://creativecommons.org/licenses/by/4.0/). Additionally, it is acknowledged that processing of the CSV station data was conducted solely by the authors.

**Author contribution.** Methodology: A.I.; I.M.; B.B. Data preparation: A.I. Data visualisation: A.I. Writing original draft: A.I; I.M. Writing review & editing: A.I.; I.M.; B.B. All authors approved the final submitted draft.

**Financial support.** The authors are grateful for the financial support provided by the Provost's Council, Trinity College Dublin for this work under the Prendergast Challenge-Based Award for the project 'Life in the Currents'.

**Competing interests.** The authors declare none.

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
