## [Reviewer Report]

Review of Igoe et al

The paper by Igoe et al concerns a clever machine learning model (XGBoost) fitted to suspended sediment concentration (SSC) data in various parts of Ireland. The main novelty seems to be the use of transfer learning to first fit the model to a large part of the data set, and then tailoring the model fit to individual coastal types (CTs). The paper is well written and the figures are informative and helpful. I do have a number of queries about the paper but I think if these were addressed it would certainly be suitable for publication. My expertise / interest is mostly in the machine learning aspect of the approach so my comments naturally focus on this area.

Queries:

In the data section, there is a lack of plots and description of the data. A few plots showing, e.g. maps of coastal types, or some of the satellite images would be really helpful to the reader. Much later in Figures 3 and 4 there are rose plots of wind direction. Are these also data? It doesn’t seem that they are results (though you could still make an argument for keeping them next to the other figures). The fact that the target variable is heavily skewed (and modelled on the log scale I think) means that a plot would surely be helpful to identify whether the variance has stabilised before the ML has been run.

The section ‘Satellite Imagery’ is quite confusing. We’re not told what ‘green’ or ‘SWIR’ are, and we’re also not told what the B categorisations are. I’m assuming that the readers of Coastal Futures might not know this (I didn’t) so I would suggest including these definitions.

In the ‘Combined dataset’ section there is a necessary discussion of the temporal relationship between the images and the ground truth, and some useful discussion. However, the reader is told that ‘This study uses an overpass of ≤2 days”. But when we get to Figure 2B, it seems like more than 2 days is used? (Orange bars - 2-7 days).

I found the ‘SSC models’ section very confusing. I think it’s just poorly written. Perhaps it would be neater to just add a table of which models were used to compare the different approaches on the same data, with a column indicating examples and strengthens/weaknesses, and references where they have been used previously. More generally, I did find the transfer learning approach a little confusing too. I think a standard data scientist would take the approach that all the data goes into the model, and then we produce the predictions. It’s not clear to me why fitting the model to a portion of the data, then fitting it again to a different set of data broken down by category, would be superior to the overall model. Perhaps it is - but I would like to see the approach compared to the full data model set up in the results figures, if possible.

I also go to confused at this point because the transfer learning section introduces some data which is called the ‘coastal type’. I cannot find the word ‘Costal Type’ used anywhere in the data section. Surely if this is part of the data (and key to the transfer learning) it should be introduced?

I was surprised to see LOO-CV used as the evaluation method. I would have guessed that this would be a very computationally challenging model to fit whilst removing every single data point. I’m assuming that all the RMSE results in Figure 2 are for the out of sample data? (This isn’t stated in the legend). I also found it confusing that there was an uncertainty bar on the RMSE in Figure 2B. If there is one predicted value per left out data point, then once a complete run of LOO is completed, you can compute one single RMSE. How were the uncertainty bands created? (Also I don’t like the vertical only bands, far better to do a box plot or violin plot of the RMSE values).

Most fundamentally, I was disappointed to find no code or data linked to the paper. At the very least a new methodological advance like this should have the code in a Git repository for people to run the model. It would be even better if the data could be included too (I can’t see anything in the paper to say that the data are not allowed to be shared).

Minor points and typos

- Abstract: “The proposed machine learning model improves RMSE by up to 71%”. Compared to what?

- The impact statement has no references? If this is by instruction then that’s fine, but it reads a little oddly without them

- L41 weird spacing before …are thus

- L182. When I read the sentence: “The primary goal of this study is to develop a model capable of distinguishing between areas with high and low SSC, rather than achieving precise accuracy at specific point locations.” I was expecting a classification model. This should probably be reworded. (L521 also states that this is classification, which it appears not to be).

- Figure 3D and E have values of SSC predicted below zero. I’m guessing this is something to do with the kernel density estimate applied to the values. It might be better to produce a histogram.

---

## [Reviewer Report]

This is an interesting paper which is highly appropriate for ‘Coastal Futures’. The application of ML methods to coastal systems is an emergent and rapidly moving field which needs papers like this one. It is generally well written although not always well argued or well structured. It is, therefore, a rather tougher read than needs to be the case. I have made extensive comments on the manuscript in the hope that it can be ‘tightened’ to give better focus on the topic under study and a better flow to the overall argument. While the Results (although it must be made clear when modelled SSC v. empirical SSC are being discussed) and Discussion are well argued and clear, but the Introduction and the RS background is sometimes seriously problematic– the claims for the need to know about SSC is over-extended in places and the RS material under-referenced. There needs to be a little more lead-in on RS methods (how do we get to surface reflectance for example). The aims of the paper do not appear until some way down page 3. This suggests to me that the preceding text is over-long and not sufficiently problem-orientated. Some of the Methods material is very dense and use terminologies which will only be meaningful to insiders of these kinds of modelling approaches. Perhaps some thought needs to be given to the use of Supplementary Material. My one substantive analytical concern is over the removal of cloud cover images which, I agree, has to be done. What is removed and what survives is not clear and is the analysis then biased by the exclusion of times of high SSC under windy, cloudy cyclonic conditions? The discussion of the acceptable lag between SSC measurements and satellite overpasses hints at this difficulty. Presentation is good although some attention is needed to the flipping between present and past tenses.

---

## [Editor Report]

Dear authors,

your paper is an interesting contribution towards the application of remote sensing data collection using ML. I agree with the reviewers that more explanation is needed regarding the satellite bands. With a littlebit of extra work your paper can be published.

---

## [Reviewer Report]

This is a thorough revision of a previous submission; indeed in many ways it might be seen as a completely new paper, given the identification of errors in the original submission and the scale of changes indicated here on the ‘track changes’ version of the re-submission. On the questions raised by the reviewers of the initial submission:

• there is now a much better lead in to the use of RS in SSC estimation

• the methods section has been extended and much improved, and with more on the data pre-processing

• there is now clear access to a GitHub repository

The paper is therefore much closer to being acceptable for publication. However, some further work is still needed.

Rather more explanation of the statement ‘GBoost was found to be the best machine learning model for remotely-sensed SSC in coastal Ireland’ is required(lines 302-303).

Under ‘Feature Importance’ it is clear that ‘Longitude’ must be a surrogate for some form of environmental control. What are these ‘regional patterns’? Later you say ‘differences in coastal geomorphology, hydrodynamics, and sediment characteristics between the Irish Sea and Atlantic-facing coasts’ but you don’t actually what those differences are. How is the reader to guess what they are? You need to say what they are.

Also, I would like to see more background information on the characteristics of Storm Agnes and Storm Lorenzo (but this was after the SSC spike?). Was their impact on SSC spikes from high rainfall (indirectly, from high runoff) or from high windspeeds? Or both? (and in what proportion). The Conclusions state ‘multi-temporal imagery of Wexford, Ireland, revealed seasonal and event-driven sediment patterns that were consistent with known meteorological, hydrodynamic, and fluvial processes at that site.’ (lines 510-513). But, like the point above, we are never told exactly what those processes were and hence the reader is asked to take this statement on trust. More explanation is needed.

The references need some attention. Journal titles need to be consistently in caps. A few references are incomplete. It is odd to see the use of ISSN numbers; it would be usual to use a DOI.

Comments were raised on the original submission by ‘Reviewer A’ on an annotated version of the manuscript. There were, for example, 8 comment boxes on the Introduction. None of these appear to have been addressed – there are no ‘track changes’ showing on the Introduction. The annotations throughout the manuscript thus still need to be addressed. Did the authors receive this annotated ms?

Minor comments (based on paper version showing track changes)

Line 178: Hu et al. (2023) is a fluvial? estuarine? setting. Give a little more detail.

Line 219: I think this deep learning text needs to be strengthened by one or two references.

Line 241: why is SHAP ‘especially effective’? Compared to?

Line 247: ‘was to develop…’

Line 309: the paper by Gorelick et al. describes the data source but it not the actual data source. Give the actual access link.

Line 313: I think it would be helpful to give the time period over which data was obtained.

Line 319: this is vague. Give the range of penetration water depths. 5 m limit?

Line 408: was performed

Line 451-452: what is the link to access the XGBoost Library? How does the reader find it?

Line 627: how is high prediction error defined?

Line 632-633: plot C is very difficult to read. ‘very tidal’ is not helpful – please state in terms of tidal range

Line 754: could we have the actual seasonal differences here

Line 807: what would be the threshold to classify a SSC level as ‘high’?

The Funding Statement needs to be completed.

---

## [Reviewer Report]

Much of the confusing parts of the paper have been removed and the analysis is now much easier to read. I’m also encouraged to see a Github repository with code in it. Having said that, some of the novelty also seems to have disappeared. There’s no mention of transfer learning any more, and all the extra analysis on coastal types has been removed. In it’s current state I’d be happy to see the paper published.

One very minor point: in the XGboost section the authors mention the use of Shap values for model interpretation. My understanding is that Shap values can be used for any ML approach, not just XGboost, and the authors should probably make this a bit clearer. My guess is that they mention it here because XGboost performs best.

---

## [Editor Report]

Please notice that there are some very minor refinements needed as indicated by reviewer 2. Otherwise, the paper is ready.

---

## [Reviewer Report]

This manuscript has been through a long and thorough review proceess and has improved at each step. The authors have now addressed all the final outstanding issues (as far as they are able) and the paper should now be accepted for publication.

---

## [Editor Report]

Dear authors, all requests by the reviewer were satisfied and we believe your manuscript is ready for publication.